# Machine Learning-Based Respiration Rate and Blood Oxygen Saturation Estimation Using Photoplethysmogram Signals

**DOI:** 10.3390/bioengineering10020167

**Published:** 2023-01-28

**Authors:** Md Nazmul Islam Shuzan, Moajjem Hossain Chowdhury, Muhammad E. H. Chowdhury, Murugappan Murugappan, Enamul Hoque Bhuiyan, Mohamed Arslane Ayari, Amith Khandakar

**Affiliations:** 1Department of Electrical, Electronic and System Engineering, Universiti Kebangsaan Malaysia, Bangi 43600, Malaysia; 2Department of Electrical Engineering, Qatar University, Doha 2713, Qatar; 3Intelligent Signal Processing (ISP) Research Lab, Department of Electronics and Communication Engineering, Kuwait College of Science and Technology, Block 4, Doha 13133, Kuwait; 4Department of Electronics and Communication Engineering, School of Engineering, Vels Institute of Sciences, Technology and Advanced Studies, Chennai 600117, Tamil Nadu, India; 5Center for Excellence for Unmanned Aerial Systems, Universiti Malaysia Perlis, Perlis 02600, Malaysia; 6BioMedical Engineering and Imaging Institute (BMEII), Icahn School of Medicine at Mount Sinai, New York, NY 10029, USA; 7Department of Civil and Architectural Engineering, Qatar University, Doha 2713, Qatar

**Keywords:** respiration rate (RR), oxygen saturation (SpO2), photoplethysmogram (PPG), feature selection algorithm, Machine Learning

## Abstract

The continuous monitoring of respiratory rate (RR) and oxygen saturation (SpO2) is crucial for patients with cardiac, pulmonary, and surgical conditions. RR and SpO2 are used to assess the effectiveness of lung medications and ventilator support. In recent studies, the use of a photoplethysmogram (PPG) has been recommended for evaluating RR and SpO2. This research presents a novel method of estimating RR and SpO2 using machine learning models that incorporate PPG signal features. A number of established methods are used to extract meaningful features from PPG. A feature selection approach was used to reduce the computational complexity and the possibility of overfitting. There were 19 models trained for both RR and SpO2 separately, from which the most appropriate regression model was selected. The Gaussian process regression model outperformed all the other models for both RR and SpO2 estimation. The mean absolute error (MAE) for RR was 0.89, while the root-mean-squared error (RMSE) was 1.41. For SpO2, the model had an RMSE of 0.98 and an MAE of 0.57. The proposed system is a state-of-the-art approach for estimating RR and SpO2 reliably from PPG. If RR and SpO2 can be consistently and effectively derived from the PPG signal, patients can monitor their RR and SpO2 at a cheaper cost and with less hassle.

## 1. Introduction

Due to the recent outbreak of a novel coronavirus known as severe acute respiratory syndrome coronavirus 2 (SARS-CoV-2), the world has experienced one of its most challenging periods in history. The disease has had a significant impact on all aspects of human existence, particularly, on the economic infrastructure and medical facilities of the world. The World Health Organization (WHO) officially recognized this condition as a pandemic on 11 March 2020. Until 7 January 2023, WHO statistics indicated that there were 662,757,682 confirmed cases of COVID-19 worldwide, with 6,702,115 fatalities [1]. There is an unprecedented pressure on society due to the current healthcare system, putting strain on the ability of healthcare institutions to provide adequate care. The easy spread of COVID-19 makes the isolation of patients essential for controlling the outbreak. Furthermore, the interaction between healthcare personnel and COVID-19 patients should be limited. Temperature, blood pressure, pulse, and respiration rate are the four most critical vital signs [2] used to assess a patient’s health. The continuous monitoring of vital signs is essential for ensuring that patients with COVID-19 receive the correct treatment and medication. By utilizing remote vital sign measurement devices, the frequency of hospital visits for patients with COVID-19 and the likelihood that a patient will be exposed to the coronavirus at home can be reduced. As a result, healthcare professionals will be able to obtain vital signs information about the patient at home.

The Food and Drug Administration (FDA) has released a comprehensive policy requiring healthcare providers to use non-invasive and remotely operated devices that measure vital signs, such as body temperature, respiratory rate, heart rate, and blood pressure [2]. Remote monitoring devices can be used by healthcare providers to monitor the vital signs of patients who are staying at home. Wearables as well as associated data loggers and graphical user interfaces (GUIs) to remotely monitor a patient’s vital signs would be extremely beneficial in such a situation. In addition to monitoring low- and medium-risk COVID-19 patients in home isolation, this system can also assist those patients in need of medical assistance but who were unable to obtain it as a result of this public health emergency.

Respiration rate (RR) is one of the most significant physiological indicators that can be used to detect abnormalities in the human body. It is one of the four primary vital signs, along with blood pressure, body temperature, and heart rate. Several recent advancements in RR estimation techniques are summarized in this section [3,4,5]. Over the past few decades, pulse oximetry has been exploited as an influential health indicator to determine the percentage of blood that is oxygenated by hemoglobin (SpO2). The pulse oximetry method is used to measure the amount of oxygen in the blood (oxygen saturation). The procedure is non-invasive, does not cause pain, and evaluates the efficiency with which oxygen is transported from the heart to other parts of the body, such as the limbs. In addition to determining whether there is sufficient oxygen in the blood, it can also be used to track the health of a person who suffers from a condition that causes low levels of oxygen in the blood. Healthy individuals have a level of oxygen saturation between 96% and 100%. Hypoxia refers to a condition in which there is a decrease in the amount of oxygen, with a blood saturation level below 90%.

A variety of techniques have been used to assess the efficacy of RR algorithms utilizing ECG and photoplethysmogram (PPG) waveforms, with the majority of them being based on PPG data. There are several issues that make it difficult to reinvestigate the performance of the described algorithms. According to [3,4], more than one hundred different methods can be utilized to determine the RR from ECG and PPG. Different strategies are developed by using new iterations of time domain RR estimation and modulation fusion methods. An algorithm presented in [6] allows PPG segments that are contaminated by movement artifacts to be automatically eliminated. This makes the method a viable option for measuring child respiration rates in a hospital emergency department. In children between the ages of 5 and 12 years, they were able to reduce the MAE to 5.2 beats per minute. A novel approach was introduced to calculate the RR of the PPG signal by combining joint sparse signal reconstruction with spectral fusion [7].

To enhance RR extraction from PPG, a smart fusion strategy based on “ensemble empirical mode decomposition (EEMD)” was presented in [8]. Using two separate datasets, EEMD was applied and evaluated to achieve an accurate RR estimate [9]. To verify the algorithm, retroactive PPG-RR computations were conducted on PPG waveforms obtained from the data warehouse and compared to the RR reference values reported [10]. A study [11] compared four “time–frequency (TF)” signal representation algorithms cascaded with a particle filter to determine which was most effective in estimating RR using variations in amplitude in the transmittance mode of PPG signals collected from the fingers. An investigation of ten individuals was described [12], which found that PPG signals generated lower RR values than accelerometers. A study [13] examined the accuracy of PPG-derived respiratory frequency measurements in different body locations during normal and deep breathing. The frequency demodulation approach was used to extract respiratory signals from the PPG data of 36 healthy participants to determine the respiration frequency by analyzing the spectral power density. In [14], Charlton’s techniques [3,4] were combined with remote PPG-based data to improve the accuracy of respiration rate estimations. A few modifications were carried out to make it suitable for remote PPG (rPPG) waveforms. By utilizing contact PPG techniques, distant PPG data can be used to estimate the respiratory rate with a mean absolute error of less than 3 bpm. According to the report, the MAE and RMSE were 3.03 and 3.69 bpm, respectively.

In [15], the authors presented their findings and the analyses of a machine learning model for SpO2 calculation based on reflectance PPG signals obtained from the finger using a customized data collection device. Using these signals, the model was able to determine SpO2. They converted the regression problem into a 20-class classification problem (1 class for each discrete value between 81% and 100%). The MAE was calculated based on the predictions of the classifier. The recommended model had an MAE of 0.5 and an accuracy of 96% with a 2% error range for SpO2 values ranging from 81% to 100%. A deep neural network (DNN) was trained using data collected during the clinical testing of a wrist-worn reflectance pulse oximeter [16]. They were aiming to achieve the requisite clinical accuracy of RMSE, which is 3.5 for SpO2 levels between 70% and 100% [17]. They also showed that the DNN using regression could not achieve the required performance with supervised learning on its own (RMSE = 4.4). DNNs trained using unsupervised methods using a collection of unlabeled PPG signals were able to achieve clinical-grade accuracy (RMSE = 2.91) after being pretrained using contrastive representation learning strategies. As a result, the DNN was able to use the signals.

A unique method was developed to estimate SpO2 using reflectance pulse oximetry, with SpO2 that was generated from transmittance PPG signals with the least amount of calibration possible. A description of this method can be found in [18]. The recommended procedure yielded an MAE of 1.81%, which is significantly less than the 2% error margin considered acceptable in clinical practice. Nevertheless, subsequent statistical research indicated that the model could be improved by adding more data and increasing the selectivity of the variables. The authors published the findings of a study of reflectance PPG signals recorded in a clinical setting from 28 patients suffering from a wide range of lung diseases. In [19], the authors developed a portable continuous Blood Oxygen Saturation Monitor (SpO2) and, for the first time, examined significant design considerations with respect to commonplace applications. According to the experimental findings, the fact that the root-mean-square error of the SpO2 estimation was only 1.8 indicated that the system was functioning properly.

Few research groups analyzed RR and SpO2 from the publicly available PPG database and used machine learning models to estimate them. Therefore, there is a potential for using machine learning models to improve RR and SpO2 estimation algorithms. Moreover, no research group has estimated RR and SpO2 simultaneously using the ML model from the publicly available PPG dataset, to the best of the authors’ knowledge.

The major contributions of this work are as follows:A unified ML system is proposed to estimate RR and SpO2 from PPG.Separate ML models are trained as RR, and SpO2 gives importance to different features.Analysis of the most important features to showcase which features predict RR or SpO2 the best.

This research is divided into four main sections. In Section 1, we present an introduction to PPG and its application for RR and SpO2 estimation, as well as a brief survey of the existing literature in this area. In Section 2, we detail the dataset, preprocessing stages, feature extraction, feature selection, evaluation matrices, and ML models. The experiment results and their significance are presented and discussed in Section 3. In the same section, the results of this work are compared to those of state-of-the-art approaches. The entire study is summarized in Section 4.

## 2. Materials and Methods

In this section, we summarize a description of the dataset and the preprocessing techniques, the different features that were extracted, the different algorithms for selecting features, and the different machine learning models that were used in this study to estimate RR and SpO2.

As shown in Figure 1, the PPG signal from the publicly available BIDMC dataset [20] was first segmented into 32 s. For 5-fold cross-validation, the PPG signals were divided into 60% training, 20% validation, and 20% test sets. A filtering process was conducted on the segmented PPG signals, followed by the removal of motion artifacts. Then, significant features were identified, and feature selection methods were employed to minimize the features’ dimensions to prevent overfitting and reduce the computation time. The chosen characteristics were employed for training, validating, and testing machine learning models. Based on the properties of PPG, the RR and SpO2 values were estimated from an unseen 20% test set per fold using the PPG properties.

### 2.1. Dataset Descriptor

The BIDMC dataset used in this study was taken from the MIMIC-II dataset [19], which is publicly available. The BIDMC dataset is a collection of electrocardiograms (ECG), photoplethysmograms (PPG), and impedance pneumography (Imp) breathing signals collected from ICU patients. The dataset includes 53 recordings of ECG, PPG, and impedance pneumography signals. Each is 8 min (total 424 min) long (sampling frequency, $f_{s}=125\;\mathrm{Hz}$), obtained from adult patients (age 19–90+ years, 32 females, RR range, 5–25 bpm, SpO2, range 84–100%) [20]. The RR values varied from 5 to 25 bpm, and the SpO2 values from 84 to 100% for different patients, as shown in Figure 2. We segmented the signals into 32 s frames with 50% overlap to ensure a sufficient number of breaths occurred so that RR and SpO2 could be computed accurately [3,4,5,20]. The breathing rate would be hampered by a shorter window, while a longer window would be impractical. We collected 32 s-long 1400 PPG segments.

### 2.2. Preprocessing

This dataset contained high-frequency noise and a limited number of motion-corrupted components in the PPG waveform. The presence of these factors can impede the process of identifying features. We filtered the PPG waveforms with a Butterworth Infinite Impulse Response (IIR) Zero Phase Filter to remove high-frequency noises. Figure 3 illustrates both the raw and the filtered PPG data, as well as the raw PPG signals without motion artifacts (MA) and high-frequency noise. A sixth-order IIR filter with a cutoff frequency of 25 Hz was created in MATLAB.

The major problem with PPG signals in real-world data acquisition is that they are frequently distorted by motion artifacts. There have been several signal-processing approaches used to eliminate motion artifacts from one-dimensional data. Recently, Variational Mode Decomposition (VMD) was utilized to remove motion artifacts from PPG waveforms. Our recent studies [21] demonstrated the effectiveness of this method. This algorithm was included in our signal processing steps in order to make the signals robust, since this dataset contained some motion-corrupted signals.

### 2.3. Feature Extraction

Several meaningful features can be extracted from a PPG signal. The features can be divided into time domain, frequency domain, and statistical domains. The systolic peak, dominant frequency, and kurtosis of the signals are examples of the time-domain, frequency-domain, and statistical features, respectively. We extracted the features in accordance with the work described in [21,22]. A better understanding of the variation within a segment can be obtained by calculating the standard deviation and variance along with the mean, as stated in [21]. This is crucial when modeling for RR, since breathing creates distortion, and modulation. Thus, these features allow for an accurate estimation of RR. A total of 107 features were extracted from the PPG waveforms. Appendix A summarize several types of features retrieved in this study.

### 2.4. Feature Selection

As a result of the application of feature selection algorithms (FSA), the dimensionality of the feature set can be reduced by limiting the number of predictors to a subset, based on a score. This is helpful in a number of ways. In the first place, a feature selection algorithm reduces the amount of training time and computational complexity required, since it minimizes the number of features. Secondly, it reduces the likelihood of overfitting. Additionally, it simplifies the deployment of the model by making it lighter. Our feature selection was based on the Feature Ranking Library (FSLib) [23], a popular MATLAB library that uses adaptive weighting and selects features based on discrimination power. Nine FSAs were used in this study. Based on empirical results, the best algorithm was selected.

“FIT A Gaussian Process Regression Model (FITRGP)”: a Gaussian regression process model is trained, which returns the predictor weights that it used. The relevance of a feature is established by computing the exponential of the negative length scales that have been learned [24,25].

“Least Absolute Shrinkage and Selection Operator (LASSO)”: a lasso model is trained which provides the best features according to its algorithm. In lasso, the variance of inference is reduced by keeping the sum of absolute model weight values below a predetermined threshold [26].

“Relief Feature Selection (ReliefF)”: ReliefF is good at estimating the importance of the function for supervised models that are distance-based and use pairwise differences between the observations to predict [27,28].

“Feature Selection with Adaptive Structure Learning (FSASL)”: FSASL is centered on linear regression. Its main limitation is the large computational cost, which increases further when dealing with high-dimension data [29].

“Unsupervised Feature Selection with Ordinal Locality (UFSOL)”: In UFSOL, a triplet-based loss function is used to ensure the ordinal localization of the actual data. This results in clustering focused on distance. By imposing an orthogonal restriction on the function projection matrix, the orthogonal base clustering is simplified. Therefore, the algorithm concurrently gathers and groups features [30].

“Laplacian Method (LM)”: LM is an unsupervised algorithm where the worth of a predictor is based on its ability to conserve the locality. This method aims to model the local geometric structure by building a closest-neighbor graph. The algorithm looks for features that respect the structure of the graph [31].

“Multi Cluster Feature Selection (mCFS)”: mCFS works well when there is a sparse eigen problem, and an L1 regularized least-square function is used to solve the optimization problem [32].

“Correlation-Based Feature Selection (CFS)”: CFS selects features in a sequential backward exclusion fashion to obtain the top features. It is an embedded process that uses Support Vector Machine (SVM) [33].

“Feature Selection Via Concave Minimization (FSV)”: in FSV, the feature selection process is included within the training of an SVM by a linear programming technique [34].

### 2.5. Machine Learning

Five distinct machine learning models with 19 variants were trained, validated, and tested using five-fold cross-validation. In each case, 60% of the 1400 recordings were used for training, 20% for validation, and 20% for testing. Once the features were extracted and selected (where applicable), the machine learning models were trained. There are five best algorithms: “Support Vector Regression (SVR),” “Gaussian Process Regression (GPR),” “Ensemble Trees,” “Linear regression,” and Decision Trees.”

Gaussian Process Regression (GPR): GPR employs the Bayesian theory to train its models. The method is most suitable for small datasets. In contrast to other supervised machine learning methods, GPRs learn a probability distribution over the full range of possibilities, rather than specific values for the parameters [24].

Ensemble Trees (ET): the ET method combines many decision trees to provide a single, unified prediction. This algorithm’s value lies in its ability to combine several weak learners into a single robust learning algorithm [35].

Support Vector Regression (SVR): the SVR method is used here to solve a regression problem using support vector machines. The SVR is trained with a symmetrical loss function, which penalizes both higher and lower mispredictions [36].

Decision Trees (DT): DT is a type of supervised learning that falls under the broad category of tree-based models. By utilizing the simple rules that were learned from the features of the data, it is able to predict the value. In this context, a tree can be viewed as a piecewise constant approximation [37].

Linear Regression (LR): the LR method involves the learning of a model with linear coefficients. To determine the optimal coefficients, the model reduces the residual sum of the squares of errors [38].

### 2.6. Evaluation Criteria

In this study, we used five different performance matrices. We will refer to *X_p_* as the projected data, *X* as the actual data, and *n* as the total number of samples or recordings.

Mean Absolute Error (MAE): the Mean Absolute Error (MAE) is the average of the absolute errors anticipated.
(1)$$\mathrm{MAE}\;=\frac{1}{n}\sum_{n}\left|X_{p}-X\right|$$

RMSE (Root-Mean-Squared Error): the RMSE computes the standard deviation of the prediction error or residuals, where the residuals are the distance between the data points and the regression line. Consequently, the RMSE quantifies the spread of the residuals, and the better the model, the narrower the spread.
(2)$$\mathrm{RMSE}\;=\sqrt{\frac{\sum {\left|X_{p}-X\right|}^{2}}{n}}$$

Correlation Coefficient (R): R is used to calculate the degree to which two variables (prediction and ground truth) are linked. It is a statistical approach that also informs us how near the forecast is to the truth.
(3)$$R\;=\sqrt{1-\frac{MSE\left(Model\right)}{MSE\left(Baseline\right)}}$$
where MSE (Baseline) = $\frac{\sum {\left|X-mean\left(X\right)\right|}^{2}}{n}$.

2SD: Standard deviation (SD) is a statistical technique that measures the spread of data relative to its mean. It is calculated by computing the square root of the variance. 2SD is the double of SD. 2SD is important because it represents the 95% confidence interval.
(4)$$2\mathrm{SD}\;=2\;\times \;\mathrm{SD}\;=2\;\sqrt{\frac{\sum error-mean{\left(error\right)}^{2}}{n}}$$
where error = $X_{p}-X$.

Limit of Agreement (LOA): the range within which a fraction of the discrepancies between two measurements (ground truth and forecast) are determined to lie. Both random (precise) and systematic occurrences are included in the LOA (bias). As a result, it is a useful technique for evaluating the effectiveness of ML models. We considered 95% of LOA for this investigation. 

## 3. Results and Discussion

This section summarizes the performance of the machine learning algorithms and the feature selection algorithms. GPR performed the best as a machine learning algorithm for both RR and SpO2. In addition, it was shown that the use of feature selection algorithms improvds the results in Table 1 and Table 2. Fitrgp and ReliefF exhibited the best performance for RR and SpO2, respectively. Table 1 and Table 2 present the results of the experiments regarding the estimation of RR and SpO2, respectively. A total of nine FSAs were used to reduce the number of features. In the tables, the results are shown for the top n features, where n is the number of features that produced the best results. A machine learning model was also trained using all the features for comparison purposes. Five different machine learning models were trained for each feature set. 

Table 1 illustrates that GPR was the most effective machine learning algorithm for all types of feature sets. It produced an RMSE of 1.45 and an MAE of 1.00 when all features were considered. There were only a few FSA and machine learning algorithms that could beat this combination. Fitrgp and GPR outperformed all other combinations. with RMSE and MAE of 1.41 and 0.89, respectively. It is important to note that only eight out of 107 features were utilized in this combination. This will simplify the deployment of the entire system.

Figure 4 illustrates the relative importance of the features selected by the Fitrgp algorithm. The algorithm selected a good blend of frequency-domain, time-domain (both PPG and derivatives of PPG), and statistical features. As a result, the maximum frequency appeared as the most important characteristic.

According to Table 2, GPR was the best machine learning algorithm for all types of feature sets. A combination of all features produced an RMSE of 1.23 and an MAE of 0.73. This combination was only surpassed by a few FSA algorithms and machine learning algorithms. ReliefF and GPR outperformed all other combinations, with RMSE and MAE values of 0.98 and 0.57, respectively. It is important to note that only 11 of 107 features were used in this combination. As a result, the deployment of the entire system will be easier.

A comparison of the relative importance of the features selected by the Relief algorithm is shown in Figure 5. The algorithm selected a good combination of frequency-domain and time-domain (PPG and derivative of PPG), as well as statistical features, similar to Fitrgp for RR. It is found that the mean of the ratio of the systolic peak time (t1) to the pulse interval (tpi) is the most important features.

Figure 6 and Figure 7 illustrate the results of the most accurate model for RR and SpO2, respectively. A regression plot and a Bland–Altman plot were used to visualize the data. The predictions were plotted against the ground truth (target) in a regression plot. In this example, two lines were drawn: one represented the ideal result in which all the targets were accurately predicted; the other represented the realistic result. A trendline of the data was also plotted. Generally speaking, the closer the trendline is to the y = x line, the better the model will be.

Using a Bland–Altman plot, the difference between the prediction and the target was plotted against the mean of the prediction and target. Using this type of visualization, it was possible to examine the spread of the data. Additionally, the 95% limit of agreement (LOA) could be detected. The smaller the LOA, the better the model, since it indicates a smaller difference between predictions and targets.

The regression plot for RR is shown in Figure 6. The R-value of the model was 0.876, which explained the distribution of the plot, as most of data were located close to the trend line. In the Bland–Altman plot, the LOA ranged from −2.795 to 2.796. Therefore, 95% of the data fell within that range. The regression plot for SpO2 is shown in Figure 7. We found an R-value of 0.951 for the model, which explained the distribution of the plot since most data were close to the trend line. The Bland–Altman plot showed an LOA from −2.036 to 2.028. This means that 95% of the data were within that range.

Studies in the literature, have several limitations, including the use of different statistical tests, the collection of data from several subject groups, and the lack of uniform implementations of algorithms, which makes it difficult to compare the stated performance of the algorithms used by the different research groups. Thus, it is impossible to determine which algorithm performs better based on the literature. In Table 3 and Table 4, RR and SpO2 estimations were compared with data from the most recent literature. Pirhonen et al. [11] proposed amplitude variations of PPG signals as a method for estimating RR. The Vortal database was used in that investigation. In this study, wavelet synchro-squeezing transformed provided the best results, with MAE and RMSE of 2.33 and 3.68 bpm, respectively. According to Jarchi et al. [12], an MAE of 2.56 bpm was obtained in a case study with ten participants in order to estimate RR from PPG signals relative to the accelerometer. Among the most effective techniques are those proposed by Motin et al. [8,9], using a smart fusion strategy based on the EEMD technology to continuously control PPG-based RR estimates. The RR estimation in daily living conditions is difficult due to motion artifacts in the PPG signal. According to their results, the MAE for MIMIC II was from 0 to 5.03, while the MAE for their own dataset was 3.05. In a recent study, L’Her et al. [10] examined the accuracy of respiratory rate measurements using a newly developed reflex-mode photoplethysmography pathological signal analysis (PPG-RR). Analyzing 30 patients in the intensive care unit (ICU), they found that the correlation coefficient for the RR was 0.78. A new method for RR estimation including motion artifact correction and machine learning models based on PPG signal characteristics was proposed by Shuzan et al. [21]. In this study, GPR and Fitrgp were found to outperform all other combinations, with RMSE, MAE, and two-standard deviation (2SD) values of 2.63, 1.97, and 5.25 breaths per minute, respectively.

In [15], an ML model was developed to calculate SpO2 values from reflectance PPG signals obtained from the finger. The SpO2 range was from 81 to 100%. A correlation coefficient of 0.95 was achieved with an LOA ranging from –2.12 to 2.12. During the signal quality index (SQI) check, approximately 30% of the total samples were lost. In [16], the authors utilized PPG datasets collected during clinical testing of a wrist-worn reflectance pulse oximeter to train a deep neural network (DNN) to achieve the requisite clinical accuracy for RMSE of 3.5%, for SpO2 levels ranging from 70% to 100% [17]. According to [39], a portable continuous Blood Oxygen Saturation (SpO2) monitor was developed, and important design concepts were examined. The RMSE of the SpO2 estimation was 1.8, which is a good estimate. The portable real-time blood oxygen saturation monitoring device reported in that study was only assessed in healthy patients with a specific SpO2 range (95–100%), which is a limitation of this study.

There is no established medical standard for estimating RR. Nonetheless, in a review study [3], over 196 conventional techniques for RR extraction were examined. They claimed that an MAE of less than 2 bpm should be considered a solid signal by a competent estimator. In Table 4, it can be seen that the machine learning model proposed in this study achieved a significantly greater degree of accuracy and precision.

Furthermore, the performance of the model proposed in this study was compared with that of a reference [17] (Table 5)**.** According to standard proposed, the MAE and RMSE of the SpO2 estimation must not exceed 2% and 3.5%, respectively. In terms of MAE and RMSE, the values obtained with the proposed model fell within the acceptable range of 0.57% and 0.98%, respectively.

This study has limitations in terms of the size of the dataset. In the BIDMC dataset, there are recordings from 53 ICU patients, which is a small sample that does not adequately represent different age groups. To train a more accurate model, eliminating this problem, a larger sample size should be considered in a future study. In addition, deep neural networks should be considered, since they learn features from signals rather than relying on handcrafted features. The performance of deep neural networks is also superior to that of classical machine learning models when the data volume is increased substantially.

## 4. Conclusions

This study proposes a novel method for estimating the respiration rate (RR) and oxygen saturation (SpO2) from photoplethysmogram (PPG) signals. In the present system, features were extracted from PPG signals, and machine learning models were used to map these features to RR and SpO2. The authors demonstrated successfully that the proposed system could estimate the RR and SpO2 values accurately compared with ground truth. For the purpose of ensuring that meaningful features were fed to the model, the authors performed pre-processing as well as feature extraction steps that had been established in previous studies. A total of 107 features were extracted, which were then reduced by using feature selection algorithms to reduce the complexity of the computations. As RR and SpO2 often have different important predictors, separate models were trained for each. A total of thirty-eight models were trained for both RR and SpO2, among which Gaussian process regression performed best. Based on the Fitrgp and ReliefF algorithms, the best features for RR and SpO2 were selected. Using the Fitrgp features, GPR provided an RMSE of 1.41 and an MAE of 0.89 when predicting RR. A GPR based on features selected by ReliefF showed an RMSE of 0.98 and an MAE of 0.57 when predicting SpO2. These models outperformed those reported in the literature and are state-of-the-art models that may be useful for developing wearable non-invasive real-time RR and SpO2 estimation devices.

## Figures and Tables

**Figure 1 bioengineering-10-00167-f001:**
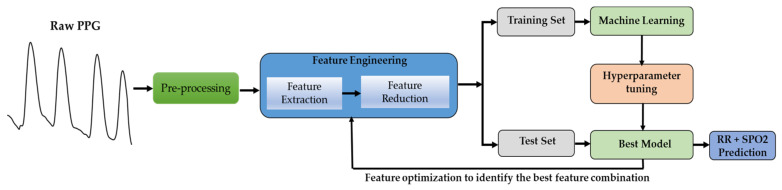
Overview of the machine learning system development.

**Figure 2 bioengineering-10-00167-f002:**
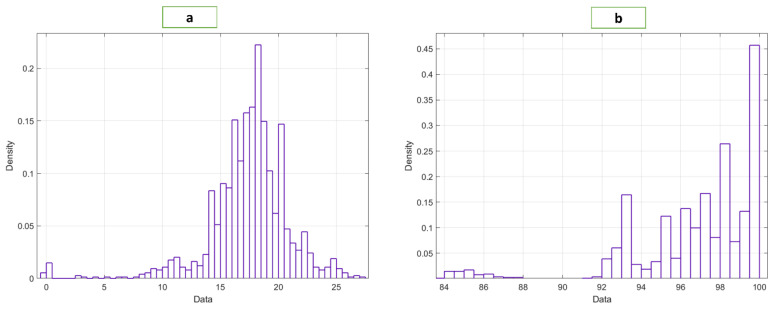
Distribution of the (**a**) RR and (**b**) SpO2 data.

**Figure 3 bioengineering-10-00167-f003:**
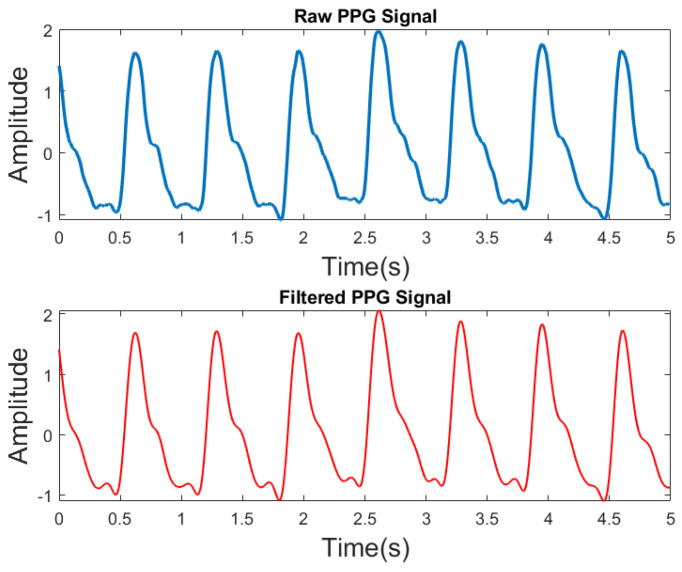
Raw PPG and filtered PPG signals.

**Figure 4 bioengineering-10-00167-f004:**
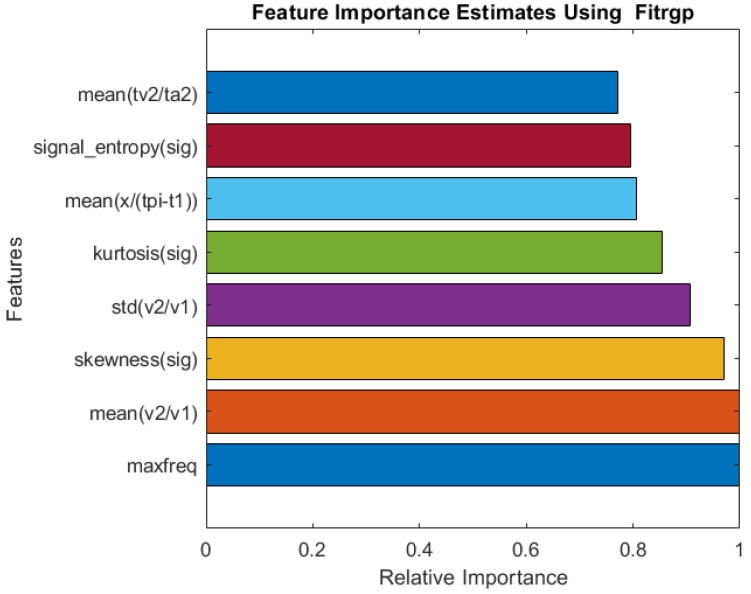
Relative importance of the top eight features selected by the Fitrgp algorithm for RR estimation.

**Figure 5 bioengineering-10-00167-f005:**
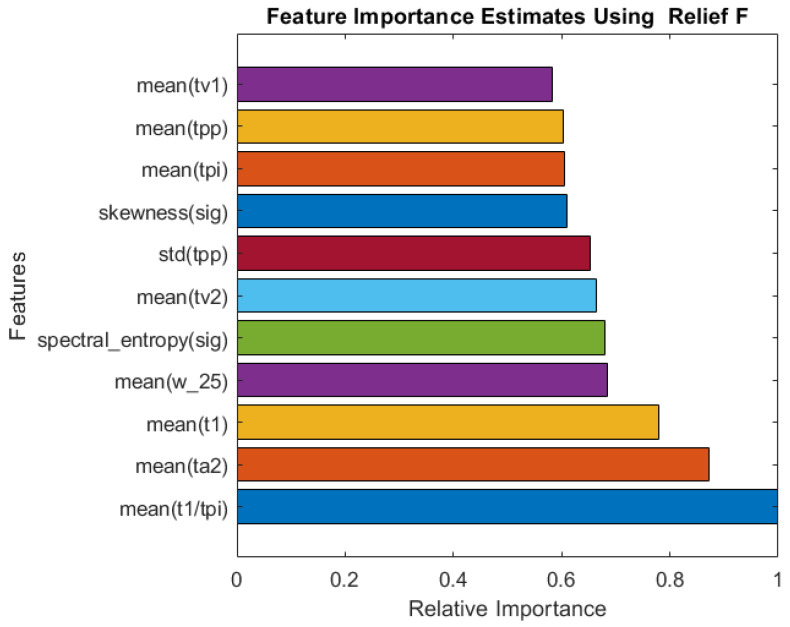
Relative importance of the top 11 features selected by the ReliefF algorithm for SpO2 estimation.

**Figure 6 bioengineering-10-00167-f006:**
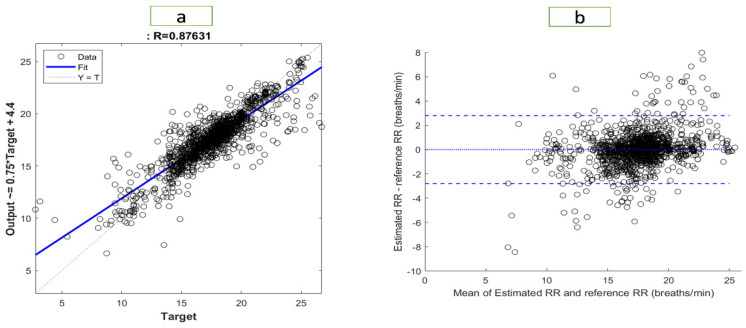
Visualization of the results for RR with (**a**) a regression plot (**b**) a Bland–Altman plot.

**Figure 7 bioengineering-10-00167-f007:**
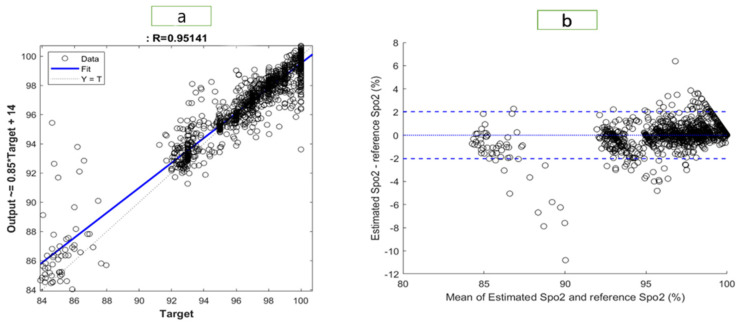
Visualization of the results for SpO2 with (**a**) a regression plot (**b**) a Bland–Altman plot.

**Table 1 bioengineering-10-00167-t001:** Evaluation of the best-performing algorithm for RR.

Selection Criteria	Top Features	Performance Criteria	GPR	Ensemble Tree	SVR	Decision Tree	Linear Regression
All Features	All	RMSE MAE	1.45 1.00	1.61 1.00	2.04 1.27	2.09 1.27	3.04 1.86
CFS	Top 28 Features	RMSE MAE	1.60 1.02	1.74 1.15	7.66 2.30	2.18 1.36	6.58 2.27
FSV	Top 30 Features	RMSE MAE	1.47 0.91	1.61 0.98	3.58 2.40	2.06 1.23	3.45 2.09
LASSO	Top 22 Features	RMSE MAE	1.43 0.90	1.69 1.06	1.98 1.19	1.98 1.23	2.63 1.99
**Fitrgp**	**Top 8 Features**	RMSE MAE	**1.41** **0.89**	1.72 1.11	1.62 0.97	2.06 1.22	2.89 2.08
ReliefF	Top 30 Features	RMSE MAE	1.51 0.99	1.66 1.04	2.10 1.26	1.94 1.19	2.71 2.04
Ufsol	Top 19 Features	RMSE MAE	1.50 0.94	1.76 1.10	1.90 1.12	2.16 1.33	2.74 2.03
Llcfs	Top 23 Features	RMSE MAE	1.57 1.04	1.82 1.14	2.00 1.20	2.27 1.40	2.86 2.15
Laplacian	Top 29 Features	RMSE MAE	1.72 1.14	1.88 1.23	2.08 1.35	2.13 1.13	10.08 2.76
Fsasl	Top 30 Features	RMSE MAE	1.80 1.09	1.96 1.29	2.19 1.44	2.24 1.21	3.80 2.05

**Table 2 bioengineering-10-00167-t002:** Evaluation of the best-performing algorithm for SpO2.

Selection Criteria	Top Features	Performance Criteria	GPR	Ensemble Tree	SVR	Decision Tree	Linear Regression
All Features	All	RMSE MAE	1.23 0.73	1.41 0.80	1.91 1.27	1.83 0.92	3.51 1.80
CFS	Top 28 Features	RMSE MAE	1.43 0.96	1.64 1.11	5.66 2.19	2.03 1.21	5.58 2.04
FSV	Top 30 Features	RMSE MAE	1.38 0.87	1.50 0.82	3.40 2.28	1.98 1.12	3.25 2.00
LASSO	Top 22 Features	RMSE MAE	1.33 0.88	1.57 0.98	1.90 1.10	1.84 1.16	2.53 1.90
Fitrgp	Top 18 Features	RMSE MAE	1.00 0.59	1.51 0.88	1.76 0.96	1.76 0.79	2.36 1.61
**ReliefF**	**Top 11 Features**	RMSE MAE	**0.98** **0.57**	1.49 1.87	1.68 0.86	1.19 0.81	3.02 2.41
Ufsol	Top 19 Features	RMSE MAE	1.39 0.90	1.61 0.99	1.77 1.01	2.06 1.21	2.62 1.97
Llcfs	Top 23 Features	RMSE MAE	1.47 0.98	1.72 1.04	1.92 1.08	2.17 1.33	2.66 2.01
Laplacian	Top 29 Features	RMSE MAE	1.62 1.08	1.78 1.11	1.97 1.25	2.03 1.05	6.80 1.67
Fsasl	Top 30 Features	RMSE MAE	1.63 0.96	1.76 1.07	2.11 1.24	2.11 1.13	3.03 1.95

**Table 3 bioengineering-10-00167-t003:** Comparison between the suggested approach and other current relevant strategies in terms of database, technique, and RR estimate error.

Author	Year	Database	Subject	Method	Metric	Result
Pirhonen et al. [11]	2018	Vortal	39 Subjects	Wavelet Synchro—squeezing Transform	MAE RMSE R 2SD	2.33 3.68 - -
Jarchi et al. [12]	2018	BIDMC	10 Subjects	Accelerometer	MAE RMSE R 2SD	2.56 - - -
Motin et al. [8]	2019	MIMIC II	53 Subjects	Empirical Mode Decomposition	MAE RMSE R 2SD	0–5.03 - - -
L’Her et al. [10]	2019	Own	30 Subjects	Own Approach	MAE RMSE R 2SD	- - 0.78 -
Motin et al. [9]	2020	Own	10 Subjects	Empirical Mode Decomposition	MAE RMSE R 2SD	3.05 - - -
Shuzan et al. [21]	2021	Vortal	39 Subjects	Machine Learning	MAE RMSE R 2SD	1.97 2.63 0.88 5.25
**This Work**	2021	BIDMC	53 Subjects	Machine Learning	MAE RMSE R 2SD	0.89 1.41 0.87 2.83

**Table 4 bioengineering-10-00167-t004:** Comparison between the suggested approach and other current relevant stragtegies in terms of database, technique, and RR estimate error.

Author	Year	Database	Subject	SpO2 Range	Method	Metric	Result
Venkat et al. [15]	2019	Own	95 subjects	81–100%	Machine Learning	MAE RMSE R 2SD LOA	- - 0.95 - −2.12 to 2.12
Priem et al. [16]	2020	Own	10 subjects	70–100%	Deep Neural Network	MAE RMSE R 2SD LOA	- 2.91 - - -
Zhang et al. [39]	2020	Own	11 subjects	95–100%	Own Approach	MAE RMSE R 2SD LOA	- 1.80 - - -
**Our Work**	2021	BIDMC	53 subjects	85–100%	Machine Learning	MAE RMSE R 2SD LOA	0.57 0.98 0.95 2.06 −2.04 to 2.03

**Table 5 bioengineering-10-00167-t005:** Comparison of this paper’s SpO2 results with those from BiOSency Bora Band.

		MAE (%)	RMSE (%)	SpO2
Standard [17]	SpO2	≤2	≤3.5	70–100%
This paper	SpO2	0.57	0.98	84–100%

## Data Availability

Not applicable.

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
