# Peer review of "Machine Learning-Based Respiration Rate and Blood Oxygen Saturation Estimation Using Photoplethysmogram Signals"

_bioengineering, 2023, doi:10.3390/bioengineering10020167_

Round 1

Reviewer 1 Report

The manuscript should be thoroughly revised regarding compliance with the requirements of the journal. I mean the structure of the article and the way the data are presented.

Conclusions. Conclusions should be based only on the results obtained. The phrase "Patients who are undergoing surgery as well as those who suffer from cardiac and pulmonary  diseases absolutely need to have their respiration rate (RR) and blood oxygen saturation level continuously monitored" does not follow from the obtained studies and should not be included in the conclusions.

The conclusions are too long and should only be based on the results obtained.

 In my opinion, the limitations of the survey conducted should be clearly written. This should be described in a separate paragraph.

Describe in detail the aim of the study and the methodology used.  The study group, the methodology used in the study should be described in detail.

Author Response

Thanks for your valuable and constructive comments and suggestions. We carefully worked on the revised paper to address all your comments and suggestions to improve its quality. 

Reviewer 2 Report

i am grateful for the opportunity to review your manuscript.

i offer the following for your consideration:

- in the third paragraph of your introduction, it is not clear what "this system" refers to. within the same paragraph, please correct "remotely" to "remote", and "will" to "would".

- at the end of your Introduction, you use the word "chunks". please use a more academically formal word.

- in section 2.4, when you introduce the FSA 'relief feature selection', 'relief' is incorrectly spelt as 'relieff'.

- in Tables 1 and 2, 'ReliefF' is incorrectly spelt as 'relieff'.

- in the paragraph between Tables 3 and 4, please re-phrase "normal people with a specific SpO2 range"; the word 'normal' should refer to the range, and not to the people.

Author Response

(The authors gave the same response as above.)

Reviewer 3 Report

(1)Please check the manuscript carefully to remove the typos, improve the language and format.

E.g.

- “signal” (in Title) Why is “s” lower case?

-The character size in Section 4 is small.

...

(2)Some background in Abstract can be shortened or condensed. The authors should highlight the contributions / originality of this article.

(3)In Section 1, the authors should clearly state the relationships between the related works and this paper. After the review, the authors should summarize the contributions of this article.

(4)Since many feature selection methods are introduced in Section 2.4, the review of the related works and comparison experiments can be more sufficient. Please carefully read, cite and compare (if applicable) the following papers (such as: Dual-source discrimination power analysis for multi-instance contactless palmprint recognition) on adaptive weighting and selection based on discrimination power. If the authors cannot employ these methods or compare their method with these methods, at least they could introduce/mention these novel technologies in related sections to improve the quality of the survey, or explain them as the possible future works.

(5)How about the performance if deep learning models are used in this paper?

Author Response

(The authors gave the same response as above.)

Reviewer 4 Report

This paper investigates several supervised learning and feature selection models to predict both respiratory rate and oxygen saturation, two important vital signs. The paper seems soundly done from the ML side of things, but needs improvement in its writing.

Major things I’m unclear about:

“the recommended model has a mean absolute error of 0.5% and an accuracy of 96” What do you mean by this? MAE is used for regression problems, accuracy for classification problems

“RMSE = 4.4%” RMSE is typically not measured in %.

“provides an average MAE of 1.81 percent,” here too, confusing

“In all, 80% of the 1400 recordings were used for training, 20% were validated, and 20% were used for testing” What do you mean?

Minor English issues: this is not an exhaustive list. Throughout the paper there are strange typos, informal language, inconsistent or incorrect capitalizations, grammatical errors, and phrases that just sound strange. The entire paper needs to be carefully checked, perhaps with the help of a native English speaker or professional service.

Be consistent with case in title. Either title case or sentence case.

“Fit a Gaussian process regression model (Fitrgp) and Relief feature” strange capitalizations

“Gaussian process regression (GPR) and the Fit a Gaussian process regression model (Fitrgp) and Relief feature selection algorithm estimate RR and SpO2 with a root mean square error (RMSE) and mean absolute error (MAE) of 1.41, 0.89 and 0.98, 0.57, respectively” not a complete sentence.

“According to statistics gathered by WHO up until Dec 13, 2023” typo

“20 March 2020” be consistent with date style

“has issued a new pol-60 icy for the health care providers” delete the

“at home or isolation unit.“->at home or in an isolation unit.

“In order to help in such situation development”->In order to help in such a situation, the development…

“There are four major chunks to this study” informal

“Infinite Impulse Response” not a proper noun, decapitalize. This goes for many other nouns in your paper. Google to see if capitalized in Wikipedia, for example.

“Finally, it makes the model more light-212 weight and hence makes it easier for deployment.” this is just a restatement of the first point.

“gaussian” capitalize

“Relieff Feature Selection”: ReliefF - inconsistent

“eigen problem” ?

Most terms like “Linear Regression” should be decapitalized.

Author Response

(The authors gave the same response as above.)

Round 2

Reviewer 3 Report

Accept.

Author Response

Reviewer 1

Comment: Accept.

Author’s Response: Thanks very much for taking the time to review the manuscript and for providing constructive suggestions in the first phase of the review. I appreciate the time the reviewer spent and the effort he put in to improve the manuscript.

Reviewer 4 Report

Thanks to the authors for their revisions. First, the authors should state the accuracy as 96% not just 96, as that is definitely a percentage. Whenever the authors pass from MAE (which is a single number for a regression problem) and accuracy (which is a percentage for a classification problem) they should make this clear.

Secondly, there are a lot of typos and English issues still remaining. The authors corrected the ones I pointed out, but I said it was not an exhaustive list. The entire paper needs to be edited carefully with the help of a native English speaker or a professional service. For example, "the globe" informal. "While there are some complex methods that uses" ->that use. "A gaussian regression" capitalize Gaussian. “Support Vector Regression (SVR),” “Gaussian Process Regression (GPR),” “Ensemble Trees,” “Linear regression, and Decision Trees.”  Most of these should be decapitalized as not proper nouns and the quotation marks are strange and not quite right eg you dont separate linear regression and decision trees.

"GPR uses Bayesian theory when it trains." Uninformative sentence.

So the paper should be acceptable once these are addressed, but this might take a lot of work. Also, the final PDF I can see is quite strange, a lot of the document is taken up by the line numbers. They take up almost half the page with this big grey box.

Author Response

Reviewer 2

Overall Comment: Thanks to the authors for their revisions. First, the authors should state the accuracy as 96%, not just 96, as that is definitely a percentage. Whenever the authors pass from MAE (which is a single number for a regression problem) and accuracy (which is a percentage for a classification problem) they should make this clear.

Secondly, there are a lot of typos and English issues still remaining. The authors corrected the ones I pointed out, but I said it was not an exhaustive list. The entire paper needs to be edited carefully with the help of a native English speaker or a professional service. For example, "the globe" informal. "While there are some complex methods that uses" ->that use. "A gaussian regression" capitalize Gaussian. “Support Vector Regression (SVR),” “Gaussian Process Regression (GPR),” “Ensemble Trees,” “Linear regression, and Decision Trees.”  Most of these should be decapitalized as not proper nouns and the quotation marks are strange and not quite right eg you dont separate linear regression and decision trees.

"GPR uses Bayesian theory when it trains." Uninformative sentence.

So the paper should be acceptable once these are addressed, but this might take a lot of work. Also, the final PDF I can see is quite strange, a lot of the document is taken up by the line numbers. They take up almost half the page with this big grey box.

Author’s Response: During our earlier revision, we noticed a number of issues with the language presentation. The incident did not happen as a result of any malicious intent. Please accept our sincere apologies. Nevertheless, we acknowledge all your concerns. and your comments and suggestions have greatly improved the manuscript. Thank you very much. A substantial revision has been made to the entire manuscript in order to improve its linguistic quality. Language or typographical errors or grammatical errors are completely eliminated from the revised manuscript.
